# HandReader: Advanced Techniques for Efficient Fingerspelling Recognition

## ABSTRACT

Fingerspelling is a significant component of Sign Language (SL), allowing the interpretation of proper names, characterized by fast hand movements during signing. Although previous works on fingerspelling recognition have focused on processing the temporal dimension of videos, there remains room for improving the accuracy of these approaches. This paper introduces HandReader, a group of three architectures designed to address the fingerspelling recognition task. HandReader$_{RGB}$ employs the novel Temporal Shift-Adaptive Module (TSAM) to process RGB features from videos of varying lengths while preserving important sequential information. HandReader$_{KP}$ is built on the proposed Temporal Pose Encoder (TPE) operated on keypoints as tensors. Such keypoints composition in a batch allows the encoder to pass them through 2D and 3D convolution layers, utilizing temporal and spatial information and accumulating keypoints coordinates. We also introduce HandReader$_{RGB+KP}$ – architecture with a joint encoder to benefit from RGB and keypoint modalities. Each HandReader model possesses distinct advantages and achieves state-of-the-art results on the ChicagoFSWild and ChicagoFSWild+ datasets. Moreover, the models demonstrate high performance on the first open dataset for Russian fingerspelling, Znaki, presented in this paper.

## 1 INTRODUCTION

Sign Language (SL) facilitates communication among more than 430 million deaf and hard-of-hearing individuals worldwide[1]. Due to the limited quantity of SL interpreters, an automatic SL Recognition (SLR) system can bridge the communication barrier between deaf individuals and people without SL knowledge(bar, 2022; 2021; bar). Such systems find application in schools, hospitals, and other social environments.

Fingerspelling is an integral component of any sign language, used for parts of speech, such as prepositions, conjunctions, and interjections, proper names, and instances without specific signs, such as addresses, abbreviations, and surnames. For example, to sign the metro station's name, the SL user should sign it letter by letter, using the corresponding signs for each letter from the SL alphabet. Therefore, fingerspelling is characterized by rapid and swift movements, complicating its recognition by blurry video frames. The motivation to tackle the fingerspelling recognition task is supported by interest from major companies, such as Google, which organized a Kaggle competition(Chow et al., 2023) to address it.

Many prior approaches to fingerspelling recognition are based on neural networks capable of processing the temporal component. To convert a video which consists of a frame sequence $I = \{I_1, I_2, ..., I_n\}$ to a letter sequence $W = \{W_1, W_2, ..W_k\}$, encoder-decoder architectures are typically utilized(Fayyazsanavi et al., 2024; Kumwilaisak et al., 2022; Shi et al., 2018). However, fingerspelling recognition, similar to Automatic Speech Recognition (ASR) and Optical Character Recognition (OCR), is complicated by the lack of alignment between $W$ and $I$, as multiple frames can interpret a single letter. The Connectionist Temporal Classification (CTC) loss(Graves, 2012) function is widely used

---

[1] https://www.who.int/news-room/fact-sheets/detail/deafness-and-hearing-loss

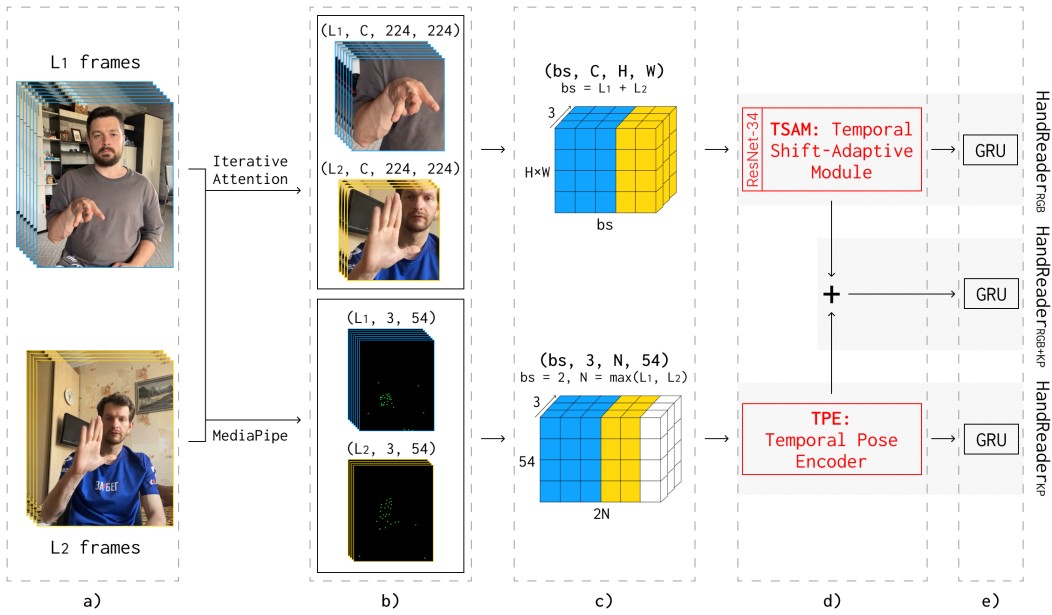

Figure 1: Three HandReader architectures. a) The videos with different lengths b) are processed to get crops and keypoints for HandReader$_{RGB}$ and HandReader$_{KP}$, respectively, c) form the batches, and d) pass through the encoders. Further, e) the GRU decoder is applied to the encoder outputs. The final output of the HandReader$_{RGB+KP}$ is obtained by summing the outputs of TSAM and TPE. Red-colored encoders are the paper's contribution.

in ASR and OCR to address this challenge by facilitating the learning of alignment between input frames and their corresponding output sequences, without the need for explicitly annotated segmentations. So, there is an option to simplify the task and enhance methods' performance by employing not only RGB frames but also such modalities as depth maps, keypoints, or combinations of video frames and keypoints(Kang et al., 2015; Shi et al., 2019; Fayyazsanavi et al., 2024; Parelli et al., 2020).

Despite multiple attempts to enhance fingerspelling recognition, the results of prior methods (Kang et al., 2015; Shi et al., 2019; Fayyazsanavi et al., 2024; Parelli et al., 2020; Shi et al., 2018; Pannattee et al., 2024; Kumwilaisak et al., 2022) on two American fingerspelling datasets – ChicagoFSWild(Shi et al., 2018) and ChicagoFSWild+(Shi et al., 2019) highlight the need for improved accuracy to ensure the applicability of existing approaches in real-world scenarios. Our work aims to enhance letter recognition accuracy while increasing frames-per-second (FPS) throughput. Such advancements can be achieved by developing novel techniques to process input data more effectively and enrich the SL domain with data annotated more precisely. However, creating fingerspelling datasets is complicated by the scarcity of people with SL knowledge and the communication barriers between them and non-SL-speaking people. To the best of our knowledge, some sign languages, such as Russian, struggle with a lack of accessible fingerspelling datasets. The Russian fingerspelling dataset is important because it is used in multinational Russia and the CIS[2] countries, such as Belarus, Ukraine, Kazakhstan, etc. This widespread comprehensibility can be explained by the historical presence of a unified sign language in these regions.

We aimed to develop an efficient approach for fingerspelling recognition and facilitate this task in the Russian SL domain with a new dataset. The contributions of this paper are summarized as follows:

---

[2]CIS – The Commonwealth of Independent States.

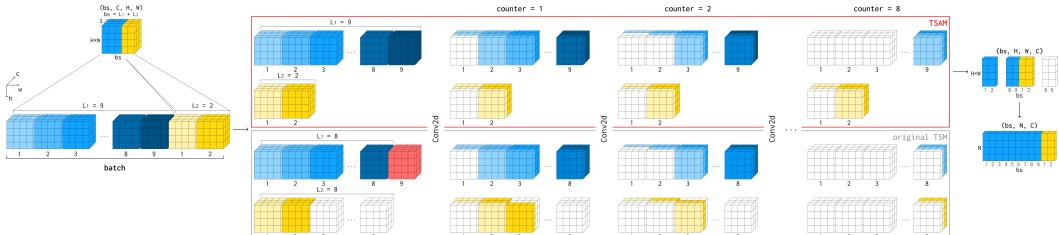

Figure 2: Comparison of the TSAM architecture (outlined in red at the top) with the original TSM (outlined in gray at the bottom). The input batch includes two videos of different lengths, with specific frames highlighted in blue and yellow. While the TSM model requires fixed-length inputs utilizing padding (shown in white) or trimming frames (shown in red) when necessary, the TSAM model employs individual shift counters to manage variable-length sequences without additional padding or trimming. The shift counter displayed at the top of the TSAM block shows how the shifts are activated individually for each video, incrementing the shift counter until it reaches the length of that specific video.

- We introduce the Temporal Shift-Adaptive Module (TSAM), extending the original Temporal Shift Module (TSM)(Lin et al., 2019) to effectively process RGB image features from videos of varying lengths without trimming or padding.
- The Temporal Pose Encoder (TPE) to operate keypoints as three-channel tensors, capturing spatial and temporal information.
- HandReader, a group of three architectures (see 1) that achieved state-of-the-art results on ChicagoFSWild and ChicagoFSWild+ test sets.
- An efficient methodology of collecting & labeling fingerspelling datasets in various languages oriented to limited signers.
- The first and the only one open dataset for Russian fingerspelling, contained 1,593 annotated phrases and over 37 thousand HD+ videos.

We release the pre-trained models, code, and dataset[3].

## 2 RELATED WORK

This overview excludes Continuous Sign Language Recognition (CSLR) research, as methods for such a task are complicated by the need to recognize the independent relationships between signs and letters within words. In contrast, fingerspelling recognition presents a more straightforward one-to-one correspondence.

Recently, the fingerspelling recognition task has been explored using various data modalities, including RGB imagery (Shi et al., 2019; 2018; Pannattee et al., 2024; Kumwilaisak et al., 2022), keypoints (Fayyazsanavi et al., 2024), and depth maps (Kang et al., 2015). Moreover, (Parelli et al., 2020) utilized data obtained by combining different modalities. Researchers are motivated to focus on the RGB modality because a fingerspelling recognition system must apply in real-world conditions and resist external changes. Meanwhile, acquiring data from other modalities requires additional equipment, such as extra cameras like Kinect for depth maps or an extra data processing stage for keypoints.

Typical videos for the fingerspelling task contain irrelevant information, such as backgrounds and foreign objects. Therefore, only RGB data requires the model to identify the most relevant areas in the frames, specifically the hands, before recognizing a sign sequence. To simplify locating the signing hand, the authors in (Shi et al., 2018) developed a hand detection method using Faster R-CNN, then trained a CNN-LSTM model for the recognition task. In their subsequent work

---

[3]https://anonymous.4open.science/r/HandReader-87FE

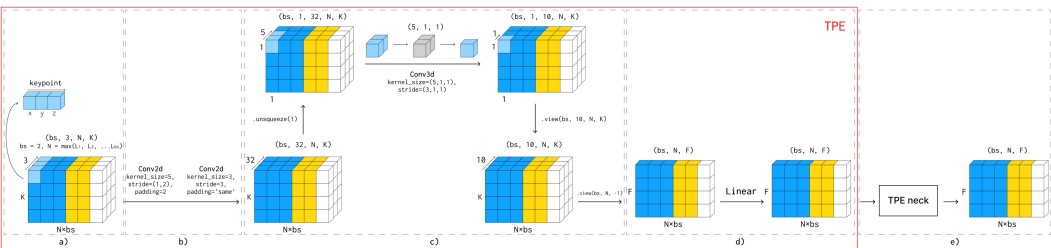

Figure 3: Demonstration of the Temporal Pose Encoder operating on a single batch. a) The encoder receives an input tensor containing $N$ frames and $K$ keypoints. All videos are padded to match the length of the most extended video clip in the batch, allowing them to be stacked. b) 2D convolutions process multiple keypoints across several frames, extracting temporal features. c) Subsequently, a 3D convolution in the form of a "tube" aggregates the extracted motion information for each body part into final feature representations $F$. d) After applying TPE, features are processed with linear layer, e) followed by MLP with convolution modules from (Gulati et al., 2020).

(Shi et al., 2019), they replaced the explicit hand detection method with an iterative attention mechanism. The authors in (Pannattee et al., 2024; Kumwilaisak et al., 2022) employed representation learning to improve the model's ability to distinguish different signs by alignment from CTC and incorporated the method from (Shi et al., 2019) as a preprocessing stage. Research (Pannattee et al., 2024) acquired state-of-the-art results on ChicagoFsWild and ChicagoFsWild+.

Since most RGB modality methods are centered around the detection hands problem, they are limited to exploring different architectures or decoding methods on the final results. In contrast, the keypoint modality avoids this issue because it includes relevant information about hand placement. The authors (Fayyazsanavi et al., 2024) trained a transformer encoder-decoder architecture with a new decoding approach, advancing performance on ChicagoFSWild and ChicagoFSWild+ test sets. Keypoints are usually acquired using MediaPipe (Lugaresi et al., 2019) or another neural network trained for pose estimation. The authors(Parelli et al., 2020) combined RGB and keypoints modalities employing linear layers to project the 2D coordinates into 3D space. After extracting all features, they concatenated them and used a decoder based on an attention mechanism. Despite the promise shown by the methods above, recognition accuracy still has potential for improvement. Therefore, our primary objective is to enhance the methodologies employed without compromising the model's weight and inference speed.

**RGB Feature Extraction in the Fingerspelling Task.** There are two main ways to extract features in the fingerspelling task. (1) Spatial features can be received by applying a 2D CNN to each frame, then extracting temporal features using 1D convolutions or RNNs. (2) Spatio-temporal features can be obtained with 3D convolution. However, processing frames as a sequence of independent frames lacks temporal information. The TSM (Lin et al., 2019) was developed to address this issue by employing 2D convolutions and shifting information along the temporal dimension. However, TSM processes fixed-size videos, which leads to the loss of valuable information in long videos or negatively affects training time and memory allocation due to the padding in shorter videos. Therefore, this paper presents a modified version of the original TSM to handle videos of varying lengths.

## 3 APPROACH

We present a group of three architectures – **HandReader** – designed to recognize sign sequences using various modalities and differ in the encoder stage only (see 1). Thus, HandReader$_{RGB}$ solely utilizes RGB features, HandReader$_{KP}$ relies on keypoints, and HandReader$_{RGB+KP}$ combines both. Each was trained directly using CTC loss, eliminating the need for frame-level character alignment.

The CTC loss accounts for all possible alignments between hand shapes and sign language letters, eliminating the need for explicit alignment supervision. More formally:

$$L_{CTC} = -log \sum_{A \in A_{X,Y}} \prod_{t=1}^{T} p_t(a_t|X)$$

where $A \in A_{X,Y}$ represents the set of valid alignments, $p_t(a_t \mid X)$ denotes the probability of symbol $a_t$ occurring at time step $t$, and $X$ is the model input.

A two-layer bidirectional Gated Recurrent Unit (GRU)(Cho et al., 2014) was chosen as a decoder to generate a sequence of characters, enhancing the model's capability to capture temporal dynamics. We apply a linear layer with SoftMax to the GRU output to generate posterior probabilities for each predicted character at every frame. Encoders for each architecture are described below.

### 3.1 HANDREADER$_{\textbf{RGB}}$

The original TSM is a video understanding approach that enables temporal modeling in 2D convolutional networks by shifting portions of channel features along the temporal dimension. However, TSM operates constrained to processing a fixed number of frames. Therefore, videos that exceed the frame limit require trimming, while shorter videos need additional empty frames (i.e., padding). Furthermore, the number of shifts is determined by the fixed frame limit. As a result, videos with extra empty frames lose critical sequential information during processing (see 2). We modified the TSM to manage a dynamic number of input frames, avoiding padding and enabling frame-by-frame processing for each video in the batch. Besides, a shift counter is introduced to preserve important sequential information, bypassing shifts on short videos.

**Temporal Shift-Adaptive Module.** The TSAM was designed to effectively process videos of various lengths. Figure2 illustrates the distinction between TSAM and TSM operations. The TSAM is incorporated into the ResNet-34 model and operates an input tensor with dimension $(bs, C, H, W)$, where $bs$ indicates batch size, equal to the sum of the video lengths, $C$ is the number of channels, $H$ and $W$ are the height and width of the frames, respectively. The TSAM also requires an auxiliary list of video lengths, denoted as $L = [L_{S_1}, L_{S_2}, \ldots, L_{S_n}]$, where $S_i$ signifies $i$th video in the batch, and $bs = \sum_i L_{S_i}$. We iterate through $L$, extracting $L_{S_i}$ frames from the batch to obtain $S_i$ video sequence and apply shifts individually, as detailed in 1. Such a method reduces VRAM usage by up to $50\%$ during batch creation, thereby facilitating the processing of longer videos.

The number of shifts is determined by the number of convolutional blocks within the backbone architecture, leading to overwrite temporal features similar to the original TSM. To prevent this loss of information, we introduce a shift counter that stops temporal shifts at the final frame, i.e., no shifting is applied if a video's length is smaller than the current shift count. Such a method ensures that each video in a batch is treated individually, resulting in more accurate feature extraction.

Each sequence of the received tensor with dimension $(bs, H, W, F)$, where $F$ represents the number of features, was subsequently padded with zeros to align with the maximum sequence length within the batch. All features are aggregated using 1D convolution, and the resulting tensor is transformed into the format $(bs, N, F)$, where $F$ is the flattened features across all channels, and $N$ is calculated as $H * W$.

### 3.2 HANDREADER$_{\textbf{KP}}$

We utilized MediaPipe(Lugaresi et al., 2019) to extract body keypoints from every frame. Face keypoints were excluded as stated in B.2, resulting in 54 $(x, y, z)$ coordinates for the pose and hand body parts per frame. The proposed feature encoder is built on a new way to structure keypoints explicitly developed for it, which organizes keypoints within tensors. Each video is treated as a tensor with dimension $(3, N, 54)$, where 54 $(x, y, z)$ vectors are stacked into matrices for each

---

**Algorithm 1** Applying temporal shifts in TSAM.

---

**Require:** Input tensor $S$ of size $(bs, C, H, W)$, list of sequence lengths $L = [L_{S_1}, L_{S_2}, \ldots, L_{S_n}]$
**Ensure:** Output tensor with temporal shifts applied
 1: **for** each sequence $S_i$ in batch from input tensor **do**
 2:     $counter \leftarrow 0$
 3:     **for** each convolutional block in backbone **do**
 4:         **if** $counter < L_{S_i}$ **then**
 5:             Apply temporal shift for sequence $S_i$
 6:             $counter \leftarrow counter + 1$
 7:         **else**
 8:             Skip temporal shift for sequence $S_i$
 9:         **end if**
10:     **end for**
11:     Apply 1D Convolution for $S_i$
12: **end for**

---

of $N$ video frames. Such a configuration enables the spatial representation of temporal data, allowing the application of processing techniques typically used for standard images. We passed the received tensor to the feature extractor described below.

**Temporal Pose Encoder.** We present a Temporal Pose Encoder based on the convolution layers. TPE operates an input tensor of dimension $(bs, 3, N, K)$, where $bs$ is the batch size, indicating the number of videos in the batch, $N$ denotes the maximum length of videos in the batch, while shorter videos were padded, and $K$ – is the amount of coordinates of keypoints. The tensor is processed by two 2D convolution layers, followed by a 3D one (see 3). 2D convolutions serve to extract both temporal and spatial features, withal transforming keypoints coordinates, enhancing the model's understanding of dynamic finger movements. At the same time, 3D convolution tries to accumulate coordinates information within a single video with the kernel-tube size of $(5, 1, 1)$, treating each coordinate independently. In this context, the hyperparameter of 5 defines the size of the receptive window along the coordinate dimension, which captures local patterns across five neighboring coordinate features. We insert a dummy dimension in the second position, enabling 3D convolution to treat the original channels as temporal dimensions. The process can be represented as follows: $(bs, 32, N, K)$ is unsqueezed to $(bs, 1, 32, N, K)$ and processed by 3D convolution to $(bs, 1, 10, N, K)$. A linear layer is then applied to project the features of the keypoints $(bs, N, F)$, where $F$ is the final encoder's features. We additionally use multi-layer perceptron (MLP) with convolution modules from (Gulati et al., 2020) as a neck between the encoder and the GRU decoder.

### 3.3 HANDREADER$_{\textbf{KP+RGB}}$

We combine features from the HandReader$_{RGB}$ and HandReader$_{KP}$ encoders to take advantage of both modalities. Two output tensors of features from TSAM and TPE are summed and passed to the GRU (see 1).

## 4 ZNAKI DATASET

We selected phrases containing proper names (1.6k unique words and expressions across 16 categories, and recruited 68 participants (37 deaf, 29 hard-of-hearing, and 2 interpreters). All candidates passed RSL and fingerspelling proficiency exams before contributing. In total, they recorded over 37k videos (42 GB) with varied environments and devices, ensuring diversity. Each video was validated by at least three annotators (deaf experts, interpreters, or RSL teachers) and automatically checked for quality.

The dataset covers all 33 Russian letters and more than 690 letter pairs, with letter frequency distributions aligned with the national language corpus. Videos average

Table 1: Open fingerspelling recognition datasets used in this work.

| Dataset | Lang | Sequences | Chars | Hours | Signers | Source |
|---|---|---|---|---|---|---|
| ChicagoFSWild | ASL | 7K | 38K | 2 | 160 | Web |
| ChicagoFSWild+ | ASL | **55K** | 0.3M | 14 | **260** | Web |
| Znaki | RSL | 37K | **0.37M** | **35** | 68 | Crowd |

Table 2: Augmentations used during training.

| Augmentatoin | Parameters | Probability | Modality |
|---|---|---|---|
| Resample | rate=(0.5, 1.5) | 0.8 | KP, RGB+KP |
| SpatialAffine | scale=(0.8, 1.2), shear=(-0.15, 0.15), shift=(-0.1, 0.1), degree=(-30, 30) | 0.75 | KP |
| TemporalMask | size=(0.2, 0.4), value=0.0 | 0.5 | KP |
| SpatialMask | size=(0.05, 0.1), value=0.0, mode="relative" | 0.5 | KP |
| HoriznotalFlip | — | 0.5 | RGB, RGB+KP, KP |
| Rotation | angle=(-10, 10) | 0.5 | RGB, RGB+KP |

102 frames, mostly in Full HD, and were split into train (58%), validation (23%), and test (19%) sets with no signer overlap. For more detailed info see 9

## 5 EXPERIMENTS

This section outlines the experimental setup and data preprocessing procedures used to assess the proposed HandReader architectures and compare them with other approaches. 1 shows the open datasets used in this work.

### 5.1 EXPERIMENTAL SETUP

All three architectures were trained from scratch using an AdamW optimizer(Loshchilov & Hutter, 2019) with an initial learning rate of 0.0001. The MultiStepLR modifies the learning rate with a gamma of 0.1 using milestones at 20 and 40 for the HandReader$_{RGB}$ and HandReader$_{RGB+KP}$ models and 25 and 40 for the HandReader$_{KP}$. We trained the HandReader$_{RGB}$ for 60 epochs, while the other two models were trained for 100 epochs each. A single Nvidia H100 with 80GB RAM was utilized for training. All models are tested on a MacBook M1 Pro using a single CPU core.

### 5.2 DATA PREPROCESSING

The ChicagoFSWild and ChicagoFSWild+ datasets are commonly used to evaluate approaches to the fingerspelling task. Since we created a new fingerspelling dataset, the evaluation results of the models trained on it are also provided.

The RGB frames for each dataset were cropped using the model described in (Shi et al., 2019), augmented with horizontal flipping and random rotation (see parameters in 2), then resized to 244 and normalized. Note that, each sign in Russian fingerspelling is invariant to horizontal flip operation. Keypoints were extracted by MediaPipe from raw frames rather than cropped images. We utilized hands and pose keypoints only, resulting in 54 keypoints. We replaced the NaN values with zeros, normalized the rest, and applied augmentations specified in 2. Only resampling, horizontal flipping, and random rotation augmentations are applied when combining two modalities in HandReader$_{RGB+KP}$.

### 5.3 RESULTS

We assessed HandReader architectures using the letter accuracy metric: $1 - \frac{S+D+I}{N}$, where $S$, $D$, and $I$ denote substitutions, deletions, and insertions, respectively, $N$ is the length of ground-truth sequence. Metrics values for other methods were obtained from the original papers for comparison purposes. Each HandReader version achieved state-of-the-art results on ChicagoFSWild and ChicagoFSWild+ test sets (see 3). Also 5 demonstrates the evaluation of presented methods on the Znaki dataset. The Znaki dataset has fewer videos than

Table 3: Results comparison on ChicagoFSWild and ChicagoFSWild+ test sets. * indicates the model was trained on both ChicagoFSWild and ChicagoFSWild+ training sets. ** means two convolution modules were used for Handreader$_{KP}$ encoder, while only one was employed for other metrics.

| Model | ChicagoFSWild | ChicagoFSWild+ | Modality |
|---|---|---|---|
| Hand Det + CNN + RNN(Shi et al., 2018) | 41.9 | 41.2 | RGB |
| Iterative Attention(Shi et al., 2019) | 45.1 | 46.7 | RGB |
| 3d hand pose + RGB(Parelli et al., 2020) | 47.9 | – | RGB + KP |
| SiamseNet+Iteraive model(Kumwilaisak et al., 2022) | 49.6 | – | RGB |
| Iter train + SupCon(Pannattee et al., 2024) | 64.9 | 71.7* | RGB |
| Pose-based transformer(Fayyazsanavi et al., 2024) | 66.3 | 71.1 | KP |
| Conformers (CFW) (Shi, 2023) | 68.4 | 60.2 | RGB |
| Conformers (CFW+CFW+) (Shi, 2023) | 77.5* | 73.0* | RGB |
| **HandReader$_{KP}$** | **69.3** | **72.4** | KP |
| **HandReader$_{RGB}$** | **72.0** | **73.8** | RGB |
| **HandReader$_{RGB+KP}$** | **72.9**** | **75.6** | RGB + KP |

Table 4: Results comparison of predictions between previous SotA solution and our models, predictions for PoseBased model were taken from the original paper.

| PoseBased(Fayyazsanavi et al., 2024)* | KP | KP+RGB | RGB | Reference |
|---|---|---|---|---|
| macah | mach | mach | mach | mach |
| meach | meah | beack | fealc | beach |
| hospital | hospital | hospital | hospital | hospital |
| - | outube | iutube | iutube | youtube |
| - | organic | organic | orgadic | organic |

ChicagoFSWild+ but includes longer, higher-resolution videos with cleaner annotations, resulting in massive margins between metrics. We also present prediction comparison between PoseBased Transformer(Fayyazsanavi et al., 2024) and our models in 3. For decoding, all models employed the beam search strategy with a beam width of 5, which explores multiple candidate sequences at each step and selects the most probable transcription.

## 6 LIMITATIONS

**Architectures.** Although the HandReader$_{KP}$ and HandReader$_{RGB}$ effectively recognize fingerspelling, extra models are required to extract keypoints and hand regions crops, respectively. Such necessities add computational overhead, decreasing inference speed and limiting the model's real-time usability, particularly on devices with limited processing power. There is an opportunity to replace an additional model with center cropping augmentation in the HandReader$_{RGB}$, provided the input focuses on the hands' region in the center of the frame. The HandReader$_{RGB+KP}$ is subject to the limits of other HandReader architectures, but it can utilize only the keypoint prediction model and obtain crops operating received points.

**Znaki Dataset.** The proposed dataset is inappropriate for fully addressing the CSLR problem since fingerspelling mostly represents proper names. Despite the historical commonality of sign languages and similarity of individual signs across CIS countries, the dynamic evolution of these languages and influence of national characteristics have resulted in Russian SL fingerspelling being used primarily within Russia, while maintaining full comprehensibility mostly among the older

Table 5: The results of HandReader$_{RGB}$, HandReader$_{KP}$, and HandReader$_{RGB+KP}$ on the Znaki test set.

| Model | Model Size (MB) | Params. (M) | Inference Time (ms) | Letter accuracy |
|---|---|---|---|---|
| HandReader$_{KP}$ | 34.7 | 9.1 | 3.9 | 92.65 |
| HandReader$_{RGB}$ | 207.7 | 54.4 | 51.4 | 92.39 |
| HandReader$_{RGB+KP}$ | 222.3 | 58.2 | 55.2 | 94.94 |

Table 6: The number of unique samples contained within each category in the Znaki dataset.

| Categories | Count | Categories | Count |
|---|---|---|---|
| Moscow metro stations | 231 | Cities world | 93 |
| Social terms | 121 | Movie titles | 92 |
| Banking terms | 107 | Sights of the world | 91 |
| Countries | 99 | Name of diseases | 90 |
| Russian Cities | 99 | Sights of the Russia | 84 |
| Authors | 98 | Professions | 75 |
| Book titles | 97 | Geographical names | 70 |
| Government institutions | 96 | Person's names | 50 |

generation. Since the dataset was collected through crowdsourcing, all participants were positioned strictly facing the camera, challenging recognition from different angles. The dataset primarily reflects the demographic composition of native RSL users, which may result in limited ethnic diversity.

## 7 ETHICAL CONSIDERATIONS

All signers provided informed consent, allowing for the processing and publication of their data in compliance with the Civil Code of the Russian Federation[4]. We fully anonymized dataset annotations, protecting the identities of contributors and ensuring confidentiality. The payment for completed tasks to crowdworkers was equivalent to the average salary of a sign language interpreter relative to the time spent. Although the Znaki is intended solely for research purposes, we acknowledge the risk of misuse, which could involve identifying individuals or enabling large-scale surveillance.

## 8 CONCLUSION

This paper presents HandReader, a suite of three distinct architectures that significantly advance the fingerspelling recognition task. The Temporal Shift-Adaptive Module and Temporal Pose Encoder were developed to capture RGB and keypoint features effectively. We achieved state-of-the-art performance on ChicagoFSWild and ChicagoFSWild+ datasets. The ablation study revealed that the proposed techniques and training pipelines are essential for achieving optimal results. We also created Znaki, the first and the one open-source heterogeneous dataset for Russian fingerspelling recognition. The presented findings offer valuable insights for enhancing fingerspelling recognition systems.

## 9 APPENDIX

## A ZNAKI DATASET

While datasets containing video spelling of single cyrillic letters(Grif & Kondratenko, 2021; Kvanchiani et al., 2024) exist, they are are unsuitable for fingerspelling recognition of Russian Sign Language (RSL) due to the lack of varied transitions between letters, the speed of signing and the naturalness & variability of surroundings during execution. Creating the dataset for this purpose involves one of three methods: recording in the studio, gathering online and TV videos, and utilizing crowdsourcing platforms. The first two methods result in the same background, low signer diversity, and inappropriate video quality, while the last requires a selection of signers and a careful record check. Using the pipeline described below, we created the dataset with the crowdsourcing platform TagMe[5].

---

[4]https://ihl-databases.icrc.org/en/national-practice/federal-law-no-152-fz-personal-data-2006

[5]https://tagme.sberdevices.ru/

### A.1 PHRASES SELECTION

Since fingerspelling is primarily used for signing proper names, only words and phrases included in this group were determined for the dataset. We consulted with deaf experts and identified the 16 most common word categories (see 6), with about 100 well-known words and phrases for each category. The phrases contain 1 to 4 proper names, with an average of 1.4 words.

### A.2 SIGNERS SELECTION

We have exclusively recorded individuals who are deaf or hard-of-hearing and who are members of the VOG (All-Russian Society of the Deaf)[6] community.

**RSL knowledge test.** Candidates were required to pass the knowledge test of Russian Sign Language (RSL), which involved recording a video with a corresponding translation of Russian annotations into RSL. Deaf experts and RSL teachers checked the assignment's correctness and decided on the candidate's admission to the project.

**Fingerspelling exam.** The candidates who passed the exam were permitted to register on the crowdsourcing platform and had to pass an additional fingerspelling exam. It included a video with RSL fingerspelling and the task of finding the correct translation into Russian from a list of words. 68 candidates out of 100 were accepted to the main tasks, passing the exam with a score of at least 85% correct answers.

### A.3 VIDEO COLLECTION

The task of video collection was accompanied by written instructions that contained the recording rules and requirements for video quality. In addition, the signers were familiarized with video, where RSL teachers demonstrate the same instruction via signs. The rules for video recording were formulated as follows: only one person in the video with the gesture-performing hand entirely in the frame is permitted, and no shaking or visual effects are allowed. Besides, the platform's verification process approved only videos with a minimum HD resolution, duration over 1 second, and frame rate over 15 frames per second. Also, the system does not pass duplicate videos. Signers have the option to re-record the video. They were allowed to record videos anywhere, making the dataset heterogeneous regarding external conditions.

### A.4 VIDEO VALIDATION

Each video was checked for the correctness of the translation from Russian to RSL fingerspelling sequences. The videos were validated by a diverse group of participants, including deaf and hard-of-hearing individuals, certified RSL interpreters, and RSL teachers and educators, all of whom confirmed their proficiency in Russian Sign Language, as well as the compliance of the video with the requirements when recording the video. Each video was reviewed by a minimum of three validators. In cases where inter-annotator agreement was insufficient, additional validators were recruited until a satisfactory level of consensus was reached. There are 82% of samples aggregated with an entire agreement of each validator, reflecting a high level of reliability in the annotations.

### A.5 TIME INTERVAL ANNOTATION

Time interval annotation was required to separate the sign sequence from other activity periods. Validated videos were processed to 30 frames per second (FPS) to bind subsequent time interval annotations before this stage. Each video was annotated by three users, further aggregating to one result by open-source AggMe framework[7], resulting in 95% videos achieved with consistency of 80%.

---

[6]All-Russian Society of the Deaf, https://voginfo.ru/
[7]https://github.com/ai-forever/aggme

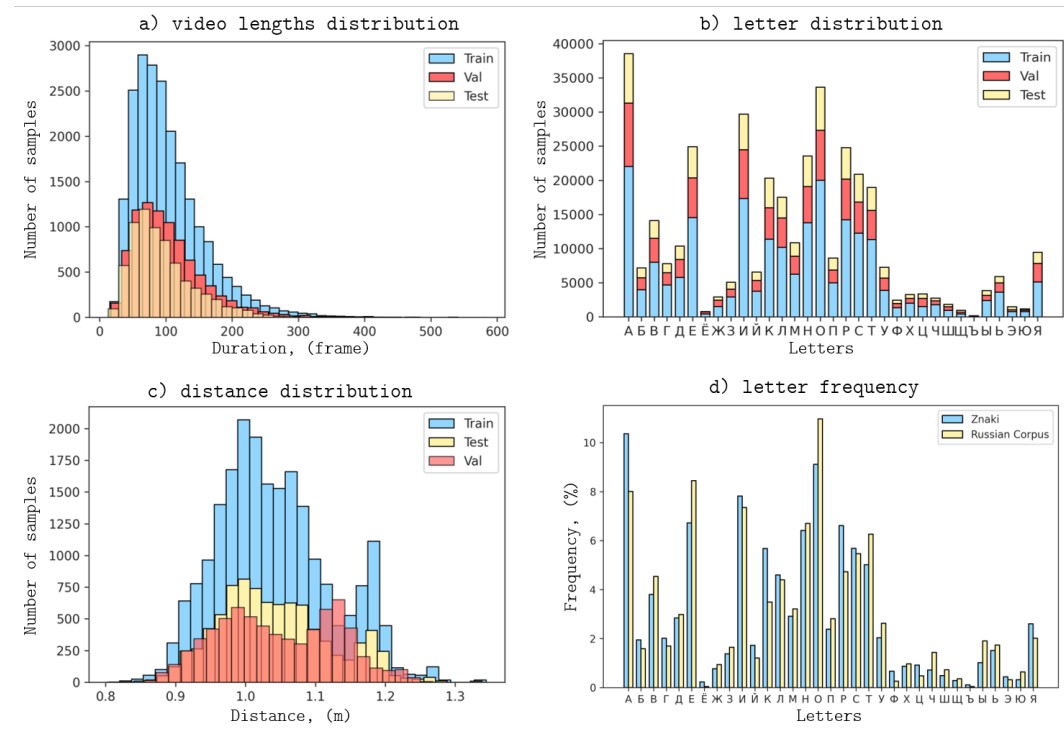

Figure 4: Znaki dataset. a) video lengths distribution; b) total number of letters distribution; c) distance distribution. Distance refers to the estimated distance between the signer and the camera: the distance is approximately estimated in meters by computed the length between the signers' left and right shoulders using MediaPipe(Lugaresi et al., 2019); d) letter frequency: comparison of letters frequency in the dataset Znaki and the Russian language corpus.

### A.6 DATASET CHARACTERISTICS

**Signers.** The dataset was recorded by 37 deaf signers, 29 hard-of-hearing, and two sign language interpreters. The signers are aged from 20 to 54 years and belong to the Caucasian race, with 52 women and 16 men. 5 provides samples from the Znaki dataset.

**Sequences.** The dataset contains 1,593 unique proper names, covering all 33 letters of the Russian alphabet and 696 pairs of different letter combinations. The distribution of letter frequency in Znaki corresponds to such distribution in the Russian language national corpus[8] (see 4b).

**Videos.** The Znaki includes 37,252 videos with a minimum frame side of 720 pixels trimmed according to time interval annotations, resulting in about 42 GB. Each sequence was recorded by between 8 and 41 different signers, averaging 23 videos per phrase. The average length of each sequence was 102 frames (see 4a). More than 80% of the dataset's video is in Full HD format, and the other is in HD. About 88% of samples were recorded on a phone camera, and the rest on a laptop.

**Split.** We split the dataset into training (58%), validation (23%), and test (19%) sets, each containing unique phrases only: 743, 300, and 550, respectively. The test set is not intersected in signers with training and validation sets.

---

[8]https://en.wikipedia.org/wiki/Russian_alphabet

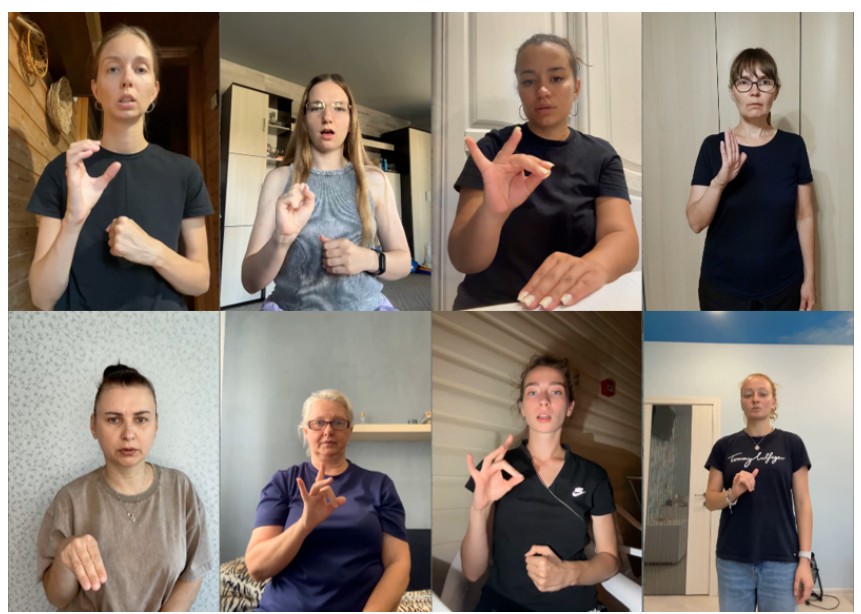

Figure 5: Sample frames from the Znaki dataset.

## B    ABLATION

This section provides an ablation study to demonstrate the impact of each component on the HandReader's overall performance. Note that HandReader$_{RGB+KP}$ was excluded from the study since it relies solely on the heuristics of the other two architectures. We maintained a consistent experimental setup described in 5 and changed only one component at a time to analyze its influence. The ChicagoF-SWild dataset was utilized in all experiments.

### B.1    HANDREADER$_{RGB}$

**Encoder.** 7 indicates that integrating TSM into the baseline model improves the performance metric by 1.69. Furthermore, replacing TSM with TSAM leads to an additional 1.4 improvement. To isolate the effect of the count shift mechanism, we implemented a variant of TSAM without the timely stopping of shifts, which led to a 1.49 performance degradation.

**Decoder.** We experimented with the LSTM (Hochreiter & Schmidhuber, 1997) decoder instead of the GRU (Cho et al., 2014) one, which resulted in a difference of 1.69. Replacing RNN with a linear layer reduces the metric by 11.0. Such observations underlined that high-level features of each frame cannot be used as independent features for final predictions.

**Others.** 7 highlights the significant impact of augmentations, as a model trained without rotations and horizontal flips reached a final metric of 70.49. Such influence is explained by the dataset containing signers with different dominant hands and videos where hands are misaligned due to camera angles or the signer's position. Additionally, we tested different batch sizes and showed that a bigger batch size is not always optimal. It is especially noticeable in ablation studies with RGB. This may happen because the batch contains videos of varying lengths, complicating the model's decoding process.

### B.2    HANDREADER$_{KP}$

**Keypoints.** We tested possible combinations of different parts of keypoints and provided results in 8. The metric was reduced by adding face keypoints since sign language users can express the same phrases differently without verbal language

Table 7: Ablation study of HandReader$_{RGB}$ components. * denotes the baseline setup with a ResNet34 configuration (without TSM or TSAM shift operations), which already outperforms previous state-of-the-art solutions. Here, we define BS as batch size and RNN as the type of recurrent neural network used, which may include LSTM, GRU, or a linear layer without recurrence. "Packed sequence" refers to a modification of the TSM mechanism that enables processing each video independently during the shift operation, as described in Section 3.1.

| Experiment | Packed sequence | RNN | Count shift | Shift type | Image Augs. | BS | Letter accuracy |
|---|---|---|---|---|---|---|---|
| best | ✓ | GRU | ✓ | TSAM | ✓ | 4 | **72.0** |
| Packed sequence | ✗ | GRU | ✗ | TSM | ✓ | 4 | $70.6_{-1.4}$ |
| RNN | ✓ | LSTM | ✓ | TSAM | ✓ | 4 | $70.31_{-1.69}$ |
| | ✓ | ✗ | ✓ | TSAM | ✓ | 4 | $61.0_{-11.0}$ |
| Count shift | ✓ | GRU | ✗ | TSAM | ✓ | 4 | $70.51_{-1.49}$ |
| Image Augs. | ✓ | GRU | ✓ | TSAM | ✗ | 4 | $70.49_{-1.51}$ |
| BS | ✓ | GRU | ✓ | TSAM | ✓ | 2 | $71.46_{-0.54}$ |
| | ✓ | GRU | ✓ | TSAM | ✓ | 6 | $70.1_{-1.9}$ |
| Baseline* | ✗ | GRU | ✗ | None | ✓ | 4 | $69.63_{-2.37}$ |

Table 8: Ablation study of HandReader$_{KP}$ components. Here, "Conv Modules" denotes the number of convolutional modules from the Conformer architecture that are used inside the encoder. Other terms are defined in 7.

| Experiment | Keypoints | | | RNN | Conv. modules | Image Augs. | BS | Letter accuracy |
|---|---|---|---|---|---|---|---|---|
| | hand | pose | face | | | | | |
| best | ✓ | ✓ | ✗ | GRU | 1 | ✓ | 4 | **69.31** |
| Keypoints | ✓ | ✗ | ✗ | | | ✓ | | $68.54_{-0.77}$ |
| | ✓ | ✓ | ✓ | | | ✓ | | $69.08_{-0.23}$ |
| | ✓ | ✗ | ✓ | GRU | 1 | ✓ | 4 | $67.59_{-1.72}$ |
| | ✗ | ✓ | ✓ | | | ✓ | | $10.02_{-59.29}$ |
| | ✗ | ✓ | ✗ | | | ✓ | | $10.36_{-58.95}$ |
| | ✗ | ✗ | ✓ | | | ✓ | | $11.08_{-58.23}$ |
| RNN | ✓ | ✓ | ✗ | LSTM | 1 | ✓ | 4 | $67.79_{-1.52}$ |
| | ✓ | ✓ | ✗ | ✗ | | ✓ | | $63.88_{-5.43}$ |
| Conv. modules | ✓ | ✓ | ✗ | GRU | 2 | ✓ | 4 | $68.09_{-1.22}$ |
| | ✓ | ✓ | ✗ | | 3 | ✓ | | $68.11_{-1.2}$ |
| Image Augs | ✓ | ✓ | ✗ | GRU | 1 | ✗ | 4 | $62.43_{-6.88}$ |
| BS | ✓ | ✓ | ✗ | | | ✓ | 2 | $69.04_{-0.27}$ |
| | ✓ | ✓ | ✗ | GRU | 1 | ✓ | 6 | $69.26_{-0.05}$ |
| | ✓ | ✓ | ✗ | | | ✓ | 8 | $68.45_{-0.86}$ |

skills. The model performs better with pose and hands keypoints than with just hands keypoints, as the pose coordinates carry information about the position of the whole hand, which may help the model better understand the hand's relative position throughout the video.

**Encoder and Decoder.** The same procedure of RNN impact assessment is provided in 8. In contrast to B.1, the usage of the linear layer does not show a significant metric drop, as the HandReader$_{KP}$ encoder contains more modules for processing temporal dimensions. We studied the impact of the number of convolution modules utilized in the encoder. As the number of convolution blocks increases, the final metric declines.

**Others.** The augmentations have a crucial effect on model performance. We also examined various batch sizes: 4, 6, and 8. As the batch size increases, there are noticeable metric drops.

### B.3 HANDREADER$_{KP+RGB}$

**Feature reduction.** We evaluated various feature reduction techniques, including simple summation, feature concatenation, element-wise multiplication, and weighted summation, obtaining 72.95, 72.95, 72.55, and 72.57 letter accuracy, respectively. We preferred the simple summation due to the smaller dimension of the sum output than concatenation one. The evaluation was conducted on the test set of the ChicagoFSWild dataset.

### B.4 DATASET'S HETEROGENEITY

We also assess the impact of user heterogeneity on the quality of the models in the Znaki dataset. We limited the training set of 743 videos in two configurations, ensuring that each phrase was represented without overlap. The first is a

Table 9: Evaluating the impact of signers heterogeneity in the training part of the Znaki dataset.

| Modilaty | Signers | Training setup | Letter accuracy |
|---|---|---|---|
| RGB | 2 | homogenous | 63,6 |
| RGB | 53 | heterogenous | **81,8**$_{+18.2}$ |
| KP | 2 | homogenous | 73,3 |
| KP | 53 | heterogenous | **83,3**$_{+10.1}$ |

Table 10: Result of comparison of predictions between different models on ChicagoFSWild dataset.

| Gt | KP | RGB | KP+RGB |
|---|---|---|---|
| youtube | outube | iutube | iutube |
| issue | isue | isue | isue |
| organic | or ganic | orgadic | organic |
| pharmacology | opharmacoint | opharoacolohy | opharmacolgi |
| pharmacology | harmcogiy | haroscothy | harmacotgy |
| physical | phisical | phylsical | phistical |
| audio | uio | ardio | atudio |
| biochemistry | biochemitry | biockemistry | biochemiry |

homogeneous training set consisting of only two signers, and the second is a heterogeneous training set – 53 signers recorded one video each, the test set was not changed. The results in 9 show that the models significantly improve on the heterogeneous set, even for the KP modality.

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
