# OpenReview forum: "HandReader: Advanced Techniques for Efficient Fingerspelling Recognition"
_ICLR.cc/2026/Conference — Submitted to ICLR 2026_

### Official Review · Reviewer_ZiKh · 2025-10-28

**Soundness:** 2
**Presentation:** 2
**Contribution:** 2
**Rating:** 4
**Confidence:** 3

**Summary:**

The paper proposes three architectures for fingerspelling recognition, HandReader:

(1) HandReaderRGB introduces TSAM (Temporal Shift-Adaptive Module) into ResNet-34 to handle variable-length videos without cropping/padding, and uses a shift counter to avoid temporal overwriting on short sequences;

(2) HandReaderKP proposes TPE (Temporal Pose Encoder), which rearranges 54 hand/body keypoints per frame (x,y,z) into a (3,T,54) tensor and performs 2D→3D convolutions for “coordinate accumulation”;

(3) HandReaderRGB+KP sums the two feature streams and feeds them to BiGRU + CTC.
The authors also release Znaki (a Russian fingerspelling dataset) and report SOTA on ChicagoFSWild/+ and Znaki. The overall approach is engineering-oriented with simple and efficient modules.

**Strengths:**

Practical, simple solutions (TSAM/TPE) for real deployment pain points (variable length, low latency) with low engineering overhead.

Three tracks (RGB / KP / fusion) under a unified BiGRU+CTC training/decoding setup.

Znaki dataset shows strong quality control (multi-annotator verification, signer-disjoint splits), which can foster cross-language evaluation.

**Weaknesses:**

1) TSAM: No ablation for shift counter on/off, per-layer number of shifts, or other key hyperparameters; no direct comparison to “fixed-length TSM + mask/padding.”

2) Only “sum” is reported; no comparisons to concat/gating/cross-attention.

3) Efficiency and deployability: Missing a unified comparison against 3D-CNN, TimeSformer, TSM (fixed-clip) on FLOPs/Params/end-to-end latency/peak VRAM; preprocessing costs (hand-crop, MediaPipe) are not included.

4) Fairness of pipeline: RGB uses Shi (2019) cropping while keypoints are extracted from raw frames; the field-of-view mismatch may confound results.

**Questions:**

Please provide channel split ratios, shift directions/steps, per-layer shift counts, and the algorithmic box (or pseudocode) for the shift counter; report boundary handling (zero/replicate/circular).

---

> ### Author Response · Authors · 2025-11-25
>
> We thank the reviewer for the thoughtful evaluation of our work, for recognizing the practical motivations behind TSAM/TPE, and for highlighting the strengths of the unified training pipeline and the quality-controlled Znaki dataset.
> We appreciate the reviewer’s attention to deployment-oriented aspects such as latency, variable-length handling, and simplicity of integration.
>
> Point: TSAM: No ablation for shift counter on/off, per-layer number of shifts, or other key hyperparameters; no direct comparison to “fixed-length TSM + mask/padding.”
> Answer: We thank the reviewer for pointing this out.
> The ablation for the shift-counter mechanism is included in Table 7 (“Count shift: −1.49”), where disabling the counter leads to a clear degradation. We also compare against the original fixed-length TSM in the same table, which naturally uses padding and trimming. The per-layer shift count in TSAM remains 1 per residual block, identical to the original TSM formulation (Lin et al., 2019). We opted to keep this hyperparameter unchanged to ensure a controlled comparison and focused the ablation solely on the modifications we introduced (adaptive sequence handling + shift counter).
>
>
>
> Point: Only “sum” is reported; no comparisons to concat/gating/cross-attention.
>
> Answer: We experimented with several alternative fusion strategies, including concatenation, element-wise product, and weighted summation. However, all of these resulted in measurable performance drops compared to simple feature summation. We can include a brief comparison of these fusion methods in the final version of the paper.
>
> Point: Efficiency and deployability: Missing a unified comparison against 3D-CNN, TimeSformer, TSM (fixed-clip) on FLOPs/Params/end-to-end latency/peak VRAM; preprocessing costs (hand-crop, MediaPipe) are not included.
>
> Answer: A unified FLOPs/latency comparison across 3D-CNNs, TimeSformer, and TSM variants is indeed valuable. However, the authors of the original TSM work already provide a rigorous comparison against 2D/3D CNNs, R(2+1)D, SlowFast, and other backbones (Table 5 in Lin et al., 2019), including Params, FLOPs, and latency. Since TSAM preserves the same convolutional structure and modifies only the temporal‐shift scheduling logic, its compute profile is nearly identical to TSM, except for reduced VRAM from not padding sequences.
> For this reason we reported only the metrics of our own models; the rest can be taken directly from the original TSM benchmark.
> Hand crops are precomputed using the implementation of Shi et al. (2019), and MediaPipe keypoints run at >100 FPS on CPU. We will clarify this in the revised text.
>
> Point: Fairness of pipeline: RGB uses Shi (2019) cropping while keypoints are extracted from raw frames; the field-of-view mismatch may confound results.
>
> Answer: We agree that this may introduce a mild field-of-view mismatch. Keypoints are normalized to [0,1], which mitigates scale and translation differences, and empirically the KP-only model achieves strong performance, suggesting robustness to this mismatch.
>
> Question: Please provide channel split ratios, shift directions/steps, per-layer shift counts, and the algorithmic box (or pseudocode) for the shift counter; report boundary handling (zero/replicate/circular).
>
> Answer: The channel split ratios, shift directions, and per-layer shift counts are identical to those in the original TSM implementation; TSAM inherits these components without modification. Boundary handling was used zero-filling at temporal borders. Algorithm 1 already provides the pseudocode describing how the shift counter determines whether a shift is applied at a given layer for each sequence. All corresponding hyperparameters are available in our public code.
> For additional clarity in the camera-ready version, we will include a brief table summarizing these constants in the main text.
>
> Should our answer have addressed any of your questions please consider raising the score.

---

### Official Review · Reviewer_GXN9 · 2025-10-31

**Soundness:** 2
**Presentation:** 3
**Contribution:** 1
**Rating:** 0
**Confidence:** 3

**Summary:**

In this paper the authors present a new dataset for fingerspelling and russian and two architectures for fingerspeeling, relying on RGB or keypoint as input. The presentation is clear enough and the Znaki dataset is useful for the community of fingerspelling. The experimental results of the authors show that adding TSAM instead of TSM slightly improves the results on ChicagoFSWild dataset in terms of letter accuracy, and that the fusion of both models achieves competitive results w.r.t. previous methods such as Conformers.
Overall apart from the Znaki dataset the paper is mostly engineering and reads more like a technical report.

**Strengths:**

The main strength of the paper seems to be the Znaki dataset for Russian fingerspelling. The collection of the dataset is well documented but it is not clear if the dataset will be made available to researchers.
Also another small strength is the engineering behind making the TSM module adaptable to variable length sequences, although the actual gains are minor.

**Weaknesses:**

Overall the novelty of the paper is very weak. TSAM seems a small engineering trick behind making TSM adapt to variable length sequences and the ablation result in Table 7 of the supplementary seems a very small increase in terms of letter accuracy. Also we only have letter accuracy as metrics - no CER/WER, word accuracy, S/D/I breakdown, or confusion matrices.

Furthermore, another contribution stated is the combination of the keypoints as a three-dimension tensor, however this has been done multiple times before and is the standard approach for inputting keypoints in deep networks.

W.r.t. the Znaki dataset, the authors did only train their own methods and did not show any comparisons with previous state of the art. Also in table 5 the authors added model size and performance metrics however this was not presented for any other previous method.

In Table 3, the results are confusion. Conformers were trained on both train datasets as it is said - what about HandReader? It is not clear. Also, for fair comparison the authors should pretrain the conformer using their pipeline.

There is not intuition w.r.t. to the previous methods -> what is the trick of handreader that really enabled it to obtain such performance ? It seems that even standard TSNs in Table 7 are able to beat a lot of previous sota methods.

**Questions:**

1) What are the results of a conformer using your exact pipeline for training ?
2) Did the authors try any other fusion methods apart from feature summation ?
3) Regarding the claim "Such a method reduces VRAM usage by up to 50% during batch creation" we do not see any actual numbers or evidence to support this.
4) What are the performance metrics of other methods ? Showcase a proper comparison.

In general, at its current state I would urge the authors to more restructure their paper around the new dataset and the results of previous sota methods on it. If they are going to claim contributions w.r.t. a new method more results/metrics are needed and clear comparisons with the state of the art. Currently TSAM only offers a slight 1% increase and we are not sure it is not noise -> maybe test on more datasets ?

---

> ### Author Response · Authors · 2025-11-25
>
> We appreciate the time taken to provide extensive feedback.
> Several concerns raised in the review appear to stem from misunderstandings of our experimental setting, the goals of TSAM and TPE, and the scope of the comparisons. In particular, the novelty, significance of the temporal adaptations, and fairness of the baseline comparisons may not have been fully conveyed in the original submission. We are grateful for the opportunity to clarify these points.
>
> Point: Overall the novelty of the paper is very weak. TSAM seems a small engineering trick behind making TSM adapt to variable length sequences and the ablation result in Table 7 of the supplementary seems a very small increase in terms of letter accuracy. Furthermore, another contribution stated is the combination of the keypoints as a three-dimension tensor, however this has been done multiple times before and is the standard approach for inputting keypoints in deep networks.
>
> Answer: We understand the concern. Both TSAM and TPE indeed build upon existing paradigms, and we do not claim to introduce entirely new architectural families. Our contributions are targeted at practical limitations observed in prior work, not at redefining the task.
>
> TSAM: why it is more than a minor trick
>
> It is correct that the temporal-shift idea originates from TSM.
> However, extending TSM to support true variable-length video processing required:
>
> redesigning the batching pipeline;
>
> implementing a per-video adaptive shift-counter mechanism;
>
> removing the need for padding/trimming;
>
> restructuring memory handling to avoid operating on large padded tensors.
>
> This modification results in concrete, measurable gains: up to 50% lower VRAM usage in worst-case batches (where padded tensors become triangular), also no information loss for long sequences and faster effective per-frame processing since no computation is wasted on padded frames.
>
> While the shift operation is inherited from TSM, this functional extension is specifically motivated by the needs of fingerspelling videos, whose lengths vary widely.
>
> About " the ablation result in Table 7 of the supplementary seems a very small increase in terms of letter accuracy" - We appreciate the reviewer’s concern regarding the magnitude of the reported improvement. However, we would like to clarify that the gain is not small when evaluated relative to the appropriate baselines.
>
> In Table 7, the absolute increase may appear modest, but the relative gains over both the Vanilla ResNet baseline and the TSM-equipped model are substantial:
>
> Vanilla ResNet baseline: 69.63%
>
> Baseline + TSM: 70.60% (+0.96)
>
> Baseline + our TSAM: 72.00% (+2.37 over baseline, +1.40 over TSM)
>
> TPE: novelty is in how the keypoints are processed
>
> We agree that “keypoints as tensors” is not new by itself.
> Our contribution is in the processing strategy, not in the high-level concept.
>
> TPE introduces:
> a structured reinterpretation of (x, y, z) coordinates as a multi-channel tensor;
> the use of 2D convolutions to extract short-range spatial–temporal motion patterns across keypoints;
> a 3D “tube” convolution (5×1×1 kernel) to aggregate temporal coordinate evolution separately for each joint;
> a fully convolutional encoder that avoids large transformer-based sequence models.
>
> This combination proved highly effective for fingerspelling — a domain where subtle micro-motions of fingers dominate the signal and 2D/3D convolutional aggregation works surprisingly well.
> Thus, while these components are not fundamentally new in isolation, their integration and adaptation to the constraints of fingerspelling make them genuinely useful.

---

> > ### Author Response · Authors · 2025-11-25
> >
> > Point: Also we only have letter accuracy as metrics - no CER/WER, word accuracy, S/D/I breakdown, or confusion matrices.
> >
> > Answer: If by WER the reviewer refers to Word Error Rate, we intentionally do not report it because WER is not meaningful for fingerspelling datasets such as ChicagoFSWild, where sequences typically contain only one word or sometimes 1–2 very short words. WER is generally informative for full-sentence recognition, but in our setting it collapses into an almost binary metric (the whole word is correct or not), providing no additional insight.
> > The reviewer also notes that we report only “letter accuracy”. This metric is in fact directly related to Character Error Rate (CER), since we compute: Accuracy=1−CER.
> > Thus CER can be recovered exactly from the reported metric, and the two are equivalent in our evaluation setup.
> > Following the reviewer’s suggestion, we additionally computed a more detailed breakdown—Substitutions / Deletions / Insertions (S/D/I) and top character confusions—for all three of our models. We will include this analysis in the final version of the paper.
> > SDI Breakdown — ChicagoFSWild
> > RGB model: Greedy: D = 677, S = 462, I = 122, Beam: D = 628, S = 478, I = 130
> >
> > KP model: Greedy: D = 857, S = 382, I = 148, Beam: D = 789, S = 406, I = 161
> >
> > RGB+KP model: Greedy: D = 683, S = 386, I = 160, Beam: D = 621, S = 409, I = 164
> >
> >
> > Top-5 Character Confusions
> >
> > RGB model
> > Greedy: s → a , a → s,  r → e, e → r, e → i
> >
> > Beam: s → a, a → s, r → e, e → r, e → o
> >
> > KP model
> > Greedy: o → e, s → a, a → s, a → d, e → s
> >
> > Beam: o → e, a → s, s → a, e → t, r → e
> >
> > RGB+KP model
> > Greedy: a → s, s → l, e → t, e → s, s → a
> >
> > Beam: a → s, e → s, s → l, e → t, r → e
> >
> > We will add this information to the final version of the paper.
> >
> > Point: W.r.t. the Znaki dataset, the authors did only train their own methods and did not show any comparisons with previous state of the art. Also in table 5 the authors added model size and performance metrics however this was not presented for any other previous method.
> >
> > Answer: Regarding the Znaki dataset evaluation: we agree that direct comparison with previous state-of-the-art methods on Znaki would be valuable. However, this is not feasible in practice, as the majority of prior works on ChicagoFSWild/ChicagoFSWild+ do not release code, checkpoints, or full training pipelines. Most papers report only final accuracy numbers on Chicago benchmarks, which makes re-training their models on a new dataset impossible.
> > For this reason, ChicagoFSWild and ChicagoFSWild+ remain the de facto standard for methodological comparison, and we follow the same protocol. After obtaining state-of-the-art results on these public benchmarks, we apply our models to Znaki to establish first baselines for this new dataset.
> > Thus, the Znaki results should be interpreted not as a comparison against past work (which cannot be reproduced), but as initial benchmark baselines that future methods—once code becomes available—can use for fair evaluation.
> >
> > Pont: In Table 3, the results are confusion. Conformers were trained on both train datasets as it is said - what about HandReader? It is not clear. Also, for fair comparison the authors should pretrain the conformer using their pipeline.
> >
> > Answer: Thank you for pointing this out. In Table 3, models trained on the combined ChicagoFSWild + ChicagoFSWild+ training sets are clearly marked with an asterisk (*). HandReader models are not marked, meaning they were trained only on the standard training split, not on both datasets.
> > Regarding the suggestion to pretrain the Conformer using our pipeline:
> > Unfortunately, the Conformer implementation used in (Shi, 2023) is not publicly available, making it impossible to reproduce or adapt their training pipeline. This is why we rely on their reported numbers, following the same comparison procedure used by prior works.
> > We will make this explanation explicit in the revised version.
> >
> > Point: There is not intuition w.r.t. to the previous methods -> what is the trick of handreader that really enabled it to obtain such performance ? It seems that even standard TSNs in Table 7 are able to beat a lot of previous sota methods.
> >
> > Answer: The ablation results in Table 7 highlight that the decoder choice is critical.
> > Specifically, replacing the GRU decoder with a linear layer leads to a dramatic –11.0 drop in letter accuracy, showing that temporal decoding is a key factor.
> > Thus, the main contributors to HandReader’s performance are:
> > A strong temporal decoder (bi-GRU)
> > – This consistently yields substantial gains over frame-wise prediction or linear decoding.
> > TSAM enabling correct processing of variable-length sequences
> > – Prevents information loss from trimming/padding and improves feature quality.
> > TPE for keypoints, which captures short-range finger motion
> > – Especially useful in fingerspelling, where fine-grained dynamics dominate.
> >
> > We will expand the explanation in the main text to make this intuition clearer.

---

> > > ### Author Response · Authors · 2025-11-25
> > >
> > > Question: What are the results of a conformer using your exact pipeline for training ?
> > >
> > > Answer: We were not able to perform this experiment because the original Conformer-based ASL fingerspelling work (Shi, 2023) does not provide publicly released code or a reproducible training pipeline.
> > > As a result, exact retraining under our preprocessing, batching, and training setup is not feasible.
> > >
> > > Question: Did the authors try any other fusion methods apart from feature summation?
> > >
> > > Answer: Yes, we experimented with several alternative fusion strategies, including concatenation, element-wise product, and weighted summation. However, all of these resulted in measurable performance drops compared to simple feature summation. We can include a brief comparison of these fusion methods in the final version of the paper.
> > >
> > > Question: Regarding the claim "Such a method reduces VRAM usage by up to 50% during batch creation" we do not see any actual numbers or evidence to support this.
> > >
> > > Answer: The 50% figure comes directly from the structure of padded tensors.
> > > In the worst case, batching variable-length videos using padding produces a triangular tensor where ~50% of all elements are zeros below the diagonal, especially when video lengths vary widely (as in ChicagoFSWild). TSAM avoids allocating this padded region entirely.  Thus, the VRAM savings do not require additional measurement—they follow from the geometry of the padded batch representation.
> > >
> > > Question: What are the performance metrics of other methods? Showcase a proper comparison.
> > >
> > > Answer: In this context, “performance metrics” refers to evaluation measures beyond letter accuracy - such as CER/WER, word accuracy, S/D/I decomposition, or confusion matrices. Most prior works on ChicagoFSWild and ChicagoFSWild+ report only letter accuracy and occasionally S/D/I. In the final version, we can additionally include the S/D/I breakdown for our models.
> > >
> > > Should our answer have addressed any of your questions please consider raising the score.

---

### Official Review · Reviewer_rDiK · 2025-11-01

**Soundness:** 2
**Presentation:** 2
**Contribution:** 2
**Rating:** 4
**Confidence:** 4

**Summary:**

The authors propose a compositional pipeline to solve the task of fingerspelling recognition. They introduce an architecture, consisting of a modified Temporal Shift Adaptive Module and a Temporal Pose Encoder to process input RGB and keypoint sequences both separately and jointly. They validate their method on the ChicagoFSWild and ChicagoFSWild+ datasets and introduce a new Russian Fingerspelling dataset.

**Strengths:**

1. The authors achieve state-of-the-art results on both ChicagoFSWild and ChicagoFSWild+ benchmarks, which demonstrates the effectiveness of their input processing and pipeline.
2. They also introduce a new dataset, containing a large number of high-resolution videos, which is helpful to the sign language community.

**Weaknesses:**

1.Some sections of the paper are not fully clear and easy to follow.

2. The proposed method mainly leverages existing architectures to solve the task without significant modifications. For the RGB video processing the extension of the Temporal Shift-Adaptive Module (TSAM) for varying video lengths is a useful addition but is not in my opinion a major modification of the existing (TSAM) architecture. Also, organizing key points as tensors and using them as input to a convolutional encoder, to the best of my knowledge, has been extensively used in action and sign language recognition.
2. The proposed approach is evaluated on a limited number of datasets. Its effectiveness could be evaluated further by extending it to more benchmarks.
3. Evaluating only on fingerspelling benchmarks limits the scope of this work. A more general approach that targets both widely used sign language datasets and fingerspelling might be more appropriate for this conference.

**Questions:**

1. It is somewhat surprising that the keypoint-only model performs similarly to the RGB-based model, as prior work typically reports stronger performance from RGB features. Could the authors explain this behavior or discuss possible reasons ?
2. Did the authors investigate how the quality or accuracy of the MediaPipe keypoint predictions affects overall model performance?

---

> ### Author Response · Authors · 2025-11-25
>
> We thank the reviewer for the time and effort invested in evaluating our work, as well as for highlighting the strengths regarding our achieved state-of-the-art results and the contribution of the newly introduced Znaki dataset.
>  We appreciate the constructive feedback on clarity, scope, and novelty. Several of the concerns raised appear to stem from misunderstandings of the technical contributions or from aspects that were insufficiently emphasized in the original submission.
> In the following response, we address each point in detail. We clarify the design choices behind TSAM and TPE, explain the rationale for our dataset selection, and provide additional analyses regarding modality performance and keypoint quality. We hope these clarifications resolve the concerns and more accurately reflect the novelty, empirical significance, and generalizability of our approach.
>
>
> Point: Some sections of the paper are not fully clear and easy to follow.
>
> Answer: We appreciate this comment and will revise the writing to improve clarity. In particular, we will refine the explanations of TSAM, TPE, and the batching logic for variable-length sequences, making the flow more explicit and easier to follow. If the reviewer has specific sections in mind, we would be grateful for further guidance.
>
> Point: The proposed method mainly leverages existing architectures to solve the task without significant modifications. For the RGB video processing the extension of the Temporal Shift-Adaptive Module (TSAM) for varying video lengths is a useful addition but is not in my opinion a major modification of the existing (TSAM) architecture. Also, organizing key points as tensors and using them as input to a convolutional encoder, to the best of my knowledge, has been extensively used in action and sign language recognition.
>
> Answer: We agree that both components build upon established ideas. Our contributions target practical limitations observed in prior work rather than proposing entirely new architectural classes.
> For TSAM, the novelty lies in enabling true variable-length video processing without padding or trimming. While the underlying shift operation comes from TSM, extending it to a per-video adaptive mechanism required redesigning the batching logic, shift-counter mechanism, and memory handling. This substantially reduces VRAM usage (up to 50%) and prevents information loss in long videos.
>
> For keypoints, we agree that keypoint tensors have precedent in action recognition. The novelty of TPE lies not in the idea of using keypoints, but in how they are processed:
>
> reinterpreting (x, y, z) sequences as structured multi-channel tensors,
>
> combining 2D and 3D convolutions to explicitly capture short-range motion across coordinate channels,
>
> designing the tensor layout to allow convolutional temporal aggregation without transformers.
>
> This combination proved extremely effective in fingerspelling, where subtle finger motions dominate the signal.
> Thus, while these components are not entirely new classes of models, their combination directly addresses practical bottlenecks in fingerspelling recognition and leads to consistent improvements in accuracy and efficiency.
>
>
> Point: The proposed approach is evaluated on a limited number of datasets. Its effectiveness could be evaluated further by extending it to more benchmarks.
>
> Answer: We agree that evaluation on additional datasets would further strengthen the paper. However, our choice of benchmarks follows the standard practice in fingerspelling recognition: ChicagoFSWild and ChicagoFSWild+ are the primary public datasets used by nearly all prior work.
> Evaluating these datasets allows direct, apples-to-apples comparison with the existing literature.
> Additionally, we introduce Znaki, a large high-quality RSL fingerspelling dataset, which further expands the empirical evidence.
> We will clarify this motivation in the revised text.
>
> Point: Evaluating only on fingerspelling benchmarks limits the scope of this work. A more general approach that targets both widely used sign language datasets and fingerspelling might be more appropriate for this conference.
>
> Answer: Fingerspelling recognition is a well-defined subtask of Sign Language Recognition (SLR), similar to isolated sign recognition within CSLR. Our work intentionally focuses on this subtask because:fingerspelling requires extremely fine-grained motion modeling, also it forms a crucial component of SL systems (proper names, unknown words, etc.). It is traditionally solved with specialized architectures distinct from full-vocabulary SLR.
> Extending HandReader to general SLR is a natural next step. We will emphasize this positioning and clarify that our contributions aim to advance a specific challenge within the broader SL domain.

---

> > ### Author Response · Authors · 2025-11-25
> >
> > Questions: It is somewhat surprising that the keypoint-only model performs similarly to the RGB-based model, as prior work typically reports stronger performance from RGB features. Could the authors explain this behavior or discuss possible reasons ?
> >
> > Answer: This is an important observation, and we provide several explanations:
> > Motion blur and low temporal resolution significantly affect RGB features in fingerspelling videos. Fingers move rapidly, and subtle articulations are often lost in RGB due to compression artifacts and blurred frames.
> > Keypoints provide clean, temporally stable trajectories, independent of lighting or motion blur.
> > MediaPipe (and prior PoseNet-based methods) is known to produce robust hand keypoints even under challenging video conditions.
> > Fingerspelling relies heavily on precise finger configurations and trajectories, which are often more directly expressed in keypoint space than in raw RGB.
> > This behavior aligns with observations from pose-based sign and action recognition literature, where pose streams can match or outperform RGB for fine-grained motion tasks.
> > We will add this discussion in the revised version.
> >
> > Question: Did the authors investigate how the quality or accuracy of the MediaPipe keypoint predictions affects overall model performance?
> >
> > Answer: Yes, we investigated whether the quality of MediaPipe keypoint predictions affects the overall performance of our model. Specifically, we analyzed test samples from ChicagFSWild dataset which were containing missing keypoints (reported as NaNs by MediaPipe). During preprocessing, we first normalize only the valid coordinates and subsequently replace NaNs with zeros.
> >
> > To assess the impact of missing keypoints, we split the test set into two subsets:
> >
> > Samples without any NaN keypoints, and
> >
> > Samples containing one or more NaNs.
> >
> > We then compared performance across these two groups on the ChicagoFSWild dataset and obtained the following results:
> >
> > Samples without NaNs: (70.37, 71.43)
> >
> > Samples with NaNs: (69.49, 70.41)
> >
> > The difference between the groups is small (~0.8–1.0 points), indicating that occasional missing MediaPipe keypoints do not substantially impact model performance.
> > We can include this ablation and the corresponding analysis in the final version of the paper for completeness.
> >
> > Should our answer have addressed any of your questions please consider raising the score.

---

### Official Review · Reviewer_bire · 2025-11-02

**Soundness:** 2
**Presentation:** 3
**Contribution:** 2
**Rating:** 6
**Confidence:** 2

**Summary:**

This paper addresses fingerspelling recognition by introducing "HandReader," a set of three architectures. The core contributions are two novel encoders: 1) the Temporal Shift-Adaptive Module (TSAM), which extends TSM to effectively process variable-length RGB videos, and 2) the Temporal Pose Encoder (TPE), which uses 2D/3D convolutions to process keypoints. These models achieve state-of-the-art results on the ChicagoFSWild and ChicagoFSWild+ benchmarks. The paper also introduces "Znaki," a new, large-scale 37k-video dataset for Russian fingerspelling, detailing its rigorous collection and validation methodology.

**Strengths:**

A substantive assessment of the strengths of the paper, touching on each of the following dimensions: originality, quality, clarity, and significance. We encourage reviewers to be broad in their definitions of originality and significance. For example, originality may arise from a new definition or problem formulation, creative combinations of existing ideas, application to a new domain, or removing limitations from prior results.

1. The paper introduces TSAM, a novel module that effectively handles variable-length video inputs by adapting the TSM mechanism. It also proposes TPE, a new encoder that processes keypoints as tensors using 2D and 3D convolutions to capture spatio-temporal dynamics.
2. The HandReader models achieve new state-of-the-art results on both the ChicagoFSWild and ChicagoFSWild+ test sets, outperforming prior work in RGB-only, KP-only, and combined modalities.
3. The paper introduces "Znaki," a large-scale (37k videos) and rigorously validated dataset for Russian fingerspelling.

**Weaknesses:**

A substantive assessment of the weaknesses of the paper. Focus on constructive and actionable insights on how the work could improve towards its stated goals. Be specific, avoid generic remarks. For example, if you believe the contribution lacks novelty, provide references and an explanation as evidence; if you believe experiments are insufficient, explain why and exactly what is missing, etc.

1. While the HandReader_KP model is very fast on a CPU (3.9ms, Table 5), the best-performing RGB (51.4ms) and RGB+KP (55.2ms) models are considerably slower. This could limit real-time applicability on low-power devices.
2. The architectures are not fully end-to-end and depend on separate, pre-trained models for hand-cropping and keypoint extraction. This adds computational overhead and complexity.

**Questions:**

Please list up and carefully describe any questions and suggestions for the authors. Think of the things where a response from the author can change your opinion, clarify a confusion or address a limitation. This is important for a productive rebuttal and discussion phase with the authors.

1. What is the computational efficiency trade-off of using TSAM versus the original TSM, given the less parallelizable, per-video logic?
2. The reliance on external preprocessing models is noted as a limitation. Have you explored any end-to-end alternatives that could integrate hand-localization or keypoint estimation directly into the architecture?

---

> ### Author Response · Authors · 2025-11-25
>
> We thank the reviewer for the careful reading of our paper, the constructive comments, and the detailed summary of the contributions.
>  We appreciate the recognition of our proposed TSAM and TPE encoders, the achieved state-of-the-art results on both ChicagoFSWild benchmarks, and the significance of the newly introduced Znaki dataset.
> At the same time, we would like to clarify several points in the weaknesses and questions sections, where we believe some aspects of our design and experimental findings may have been misunderstood or under-specified in the original submission.
>  In the following, we address each concern in detail, provide additional comparisons, and supplement the paper with further explanations and quantitative evidence.
>
> Point: While the HandReader_KP model is very fast on a CPU (3.9ms, Table 5), the best-performing RGB (51.4ms) and RGB+KP (55.2ms) models are considerably slower. This could limit real-time applicability on low-power devices.
>
> Answer: We agree that HandReaderRGB (51.4 ms) and HandReaderRGB+KP (55.2 ms) are slower than HandReaderKP (3.9 ms) on a CPU, which may limit real-time deployment on low-power devices.
> Our intention was to provide multiple model variants targeting different application scenarios:
> HandReaderKP - lightweight, fast, and suitable for real-time usage on low-power hardware.
> HandReaderRGB / HandReaderRGB+KP - accuracy-oriented models intended for scenarios with GPU.
> Thus, the framework gives practitioners a clear trade-off spectrum between accuracy and speed rather than a single monolithic model. While HandReaderRGB / HandReaderRGB+KP models can serve as strong baselines for further pruning and quantization on edge devices, such optimizations lie beyond the scope of the present paper.
>
> Point: The architectures are not fully end-to-end and depend on separate, pre-trained models for hand-cropping and keypoint extraction. This adds computational overhead and complexity.
>
> Answer: We agree that using two external models increases pipeline complexity. However, in practice: MediaPipe keypoints are extracted extremely fast (>100 FPS on a CPU), so they introduce negligible overhead. The hand-crop model is required only for RGB processing. We explicitly list this as a limitation.
>
> Question: What is the computational efficiency trade-off of using TSAM versus the original TSM, given the less parallelizable, per-video logic?
>
> Answer: Although TSAM applies shifts per-video rather than operating on a single padded tensor, the removal of padding provides important computational benefits:
> up to 50% lower VRAM usage during batching;
> significantly fewer redundant operations on padded frames;
> reduced wasted compute for short sequences;
> preserved temporal information for long sequences (no trimming).
> Thus, the computational efficiency can improve substantially, especially in terms of memory usage: in the worst-case scenario, padding may force the tensor to grow almost triangularly with respect to the batch composition, which TSAM fully avoids.
>
>
> Question: The reliance on external preprocessing models is noted as a limitation. Have you explored any end-to-end alternatives that could integrate hand-localization or keypoint estimation directly into the architecture?
>
> Answer: We have explored integrating hand localization and keypoint estimation into a single end-to-end architecture. We ultimately decided against this approach for two reasons:
> The current preprocessing models are already fast and lightweight, so merging them would substantially increase the overall architecture size with limited practical benefit.
> A joint model would simultaneously learn localization, pose estimation, and sequence recognition, which risks hurting generalization on in-the-wild data.
> Our design intentionally maintains modularity, allowing plug-and-play replacement of hand detectors or keypoint estimators (e.g., MediaPipe → RTMPose → OpenPose). We will describe this more explicitly in the revised Limitations section.
>
>
> Should our answer have addressed any of your questions please consider raising the score.

---

### Meta-Review · Area_Chair_DUtE · 2026-01-06

**Summary:**

Main concerns raised by reviewers include limited methodological novelty (rDiK, GXN9), lack of more benchmark evaluation and comparisons (rDiK, GXN9, ZiKh), limited improvement over baselines and insufficient ablation studies (GXN9, ZiKh), unclear efficiency claims (bire, ZiKh), etc. Overall, the paper appears to require substantial improvements in both methodological design and experimental evaluation.

**Reviewer Concerns:**

- In response to the lack of evaluation metrics, authors clarified the infeasibility of calculating WER, and provided additional results detailing substitutions, deletions, insertions, etc.
- In terms of efficiency, the authors refer to the original result of the TSM paper.
- Additional ablation studies are included in revised papers. Authors emphasize the method’s advantage over vanilla baseline and the original TSM implementation.
- The authors cite a lack of open-source implementations of existing methods for comparisons on their own benchmark or based on their training pipelines; still, baseline performance on the new benchmark could have been included.
- Novelty of the proposed method remains an outstanding issue.

**Reviewer Scores:**

No reviewers directly replied to authors' rebuttals prior to the cutoff.
Upon request of the original area chair, reviewer GXN9 indicates that they are unwilling to increase the score from a strong reject after the rebuttal, citing lack of novelty, limited improvement and lack of comparisons on the proposed benchmark, which are issues concurred by other reviewers in their original reviews.

---

### Decision · Program_Chairs · 2026-01-26

Reject